# Revisiting ResNets: Improved Training and Scaling Strategies

**Irwan Bello**
Google Brain

**William Fedus**
Google Brain

**Xianzhi Du**
Google Brain

**Ekin D. Cubuk**
Google Brain

**Aravind Srinivas**
UC Berkeley

**Tsung-Yi Lin**
Google Brain

**Jonathon Shlens**
Google Brain

**Barret Zoph**
Google Brain

## Abstract

Novel computer vision architectures monopolize the spotlight, but the impact of the model architecture is often conflated with simultaneous changes to training methodology and scaling strategies. Our work revisits the canonical ResNet [13] and studies these three aspects in an effort to disentangle them. Perhaps surprisingly, we find that training and scaling strategies may matter more than architectural changes, and further, that the resulting ResNets match recent state-of-the-art models. We show that the best performing scaling strategy depends on the training regime and offer two new scaling strategies: (1) scale model depth in regimes where overfitting can occur (width scaling is preferable otherwise); (2) increase image resolution more slowly than previously recommended [55]. Using improved training and scaling strategies, we design a family of ResNet architectures, ResNet-RS, which are 1.7x - 2.7x faster than EfficientNets on TPUs, while achieving similar accuracies on ImageNet. In a large-scale semi-supervised learning setup, ResNet-RS achieves 86.2% top-1 ImageNet accuracy, while being 4.7x faster than EfficientNet-NoisyStudent. The training techniques improve transfer performance on a suite of downstream tasks (rivaling state-of-the-art self-supervised algorithms) and extend to video classification on Kinetics-400. We recommend practitioners use these simple revised ResNets as baselines for future research.

## 1 Introduction

The performance of a vision model is a product of the architecture, training methods and scaling strategy. Novel architectures underlie many advances, but are often simultaneously introduced with other critical – and less publicized – changes in the details of the training methodology and hyper-parameters. Additionally, new architectures enhanced by modern training methods are sometimes compared to older architectures with dated training methods (e.g. ResNet-50 with ImageNet Top-1 accuracy of 76.5% [13]). Our work addresses these issues and empirically studies the impact of *training methods* and *scaling strategies* on the popular ResNet architecture [13].

We survey the modern training and regularization techniques widely in use today and apply them to ResNets (Figure 1). In the process, we encounter interactions between training methods and show a benefit of reducing weight decay values when used in tandem with other regularization techniques. An additive study of training methods in Table 1 reveals the significant impact of these decisions: a

Correspondence to Irwan Bello and Barret Zoph {ibello,barretzoph}@google.com. Code and checkpoints available in TensorFlow: https://github.com/tensorflow/models/tree/master/official/vision/beta and https://github.com/tensorflow/tpu/tree/master/models/official/resnet/resnet_rs

35th Conference on Neural Information Processing Systems (NeurIPS 2021).

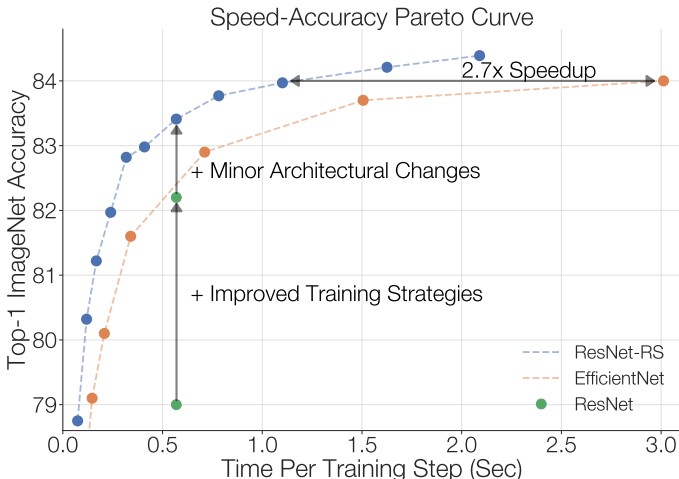

Figure 1: **Improving ResNets to state-of-the-art performance.** We improve on the canonical ResNet [13] with modern training methods (as also used in EfficientNets [55]), minor architectural changes and improved scaling strategies. The resulting models, **ResNet-RS**, outperform EfficientNets on the speed-accuracy Pareto curve with speed-ups ranging from **1.7x - 2.7x** on TPUs and **2.1x - 3.3x** on GPUs. ResNet (●) is a ResNet-200 trained at 256×256 resolution. Training times reported on TPUs.

canonical ResNet-200 with 79.0% top-1 ImageNet accuracy is improved to 82.2% (+3.2%) through *improved training methods alone*. This is increased further to 83.4% by two small and commonly used architectural improvements: ResNet-D [15] and Squeeze-and-Excitation [21]. Figure 1 traces this refinement over the starting ResNet in a speed-accuracy Pareto curve.

We offer new perspectives and practical advice on scaling vision architectures. While prior works extrapolate scaling rules from small models [55] or from short training duration [39], we design scaling strategies by exhaustively training models across a variety of scales for the full training duration (e.g. 350 epochs instead of 10 epochs). In doing so, we uncover strong dependencies between the best performing scaling strategy and the training regime (e.g. number of epochs, model size, dataset size). These dependencies are missed in any of these smaller regimes, leading to sub-optimal scaling decisions. Our analysis leads to new *scaling strategies* summarized as **(1)** scale the model depth when overfitting can occur (scaling the width is preferable otherwise) and **(2)** scale the image resolution more slowly than prior works [55].

Using the improved training and scaling strategies, we design a family of re-scaled ResNets, *ResNet-RS*, across model various scales (Figure 1). ResNet-RS models use less memory during training and are **1.7x - 2.7x** faster on TPUs (**2.1x - 3.3x** faster on GPUs) than the popular EfficientNets on the speed-accuracy Pareto curve. In a large-scale semi-supervised learning setup, ResNet-RS obtains a **4.7x** training speed-up on TPUs (**5.5x** on GPUs) over EfficientNet-B5 when co-trained on ImageNet [30] and an additional 130M pseudo-labeled images.

Finally, we conclude with a suite of experiments testing the generality of the improved training and scaling strategies. We first demonstrate that our scaling strategy improves the speed-accuracy Pareto curve of EfficientNet. Next, we show that the improved training strategies yield representations that rival or outperform those from self-supervised algorithms (SimCLR and SimCLRv2 [4, 5]) on a suite of downstream tasks. The improved training strategies also extend to video classification, yielding an improvement from 73.4% to 77.4% (+4.0%) on the Kinetics-400 dataset.

Through combining lightweight architectural changes (used since 2018) and improved training and scaling strategies, we discover the ResNet architecture sets a state-of-the-art baseline for vision research. This finding highlights the importance of teasing apart each of these factors in order to understand what architectures perform better than others. We summarize our contributions:

- An empirical study of regularization techniques and their interplay, which leads to a training strategy that achieves strong performance (e.g. +3.2% top-1 ImageNet accuracy, +4.0% top-1 Kinetics-400 accuracy) *without having to change the model architecture*.

- An empirical study of scaling which uncovers strong dependencies between training and the best performing scaling strategy. We propose a simple scaling strategy: (1) scale depth when overfitting can occur (scaling width can be preferable otherwise) and (2) scale the image resolution more slowly than prior works [55]. This scaling strategy improves the speed-accuracy Pareto curve of both ResNets and EfficientNets.

- **ResNet-RS**: a Pareto curve of ResNet architectures that are **1.7x - 2.7x** faster than EfficientNets on TPUs (**2.1x - 3.3x** on GPUs) by applying the training and scaling strategies. Semi-supervised training of ResNet-RS with an additional 130M pseudo-labeled images achieves 86.2% top-1 ImageNet accuracy, while being **4.7x** faster on TPUs (**5.5x** on GPUs) than the corresponding EfficientNet-NoisyStudent [57].

- Empirically show that representations obtained from supervised learning using modern training techniques rival or outperform state-of-the-art self-supervised representations (SimCLR [4], SimCLRv2 [5]) on suite of downstream computer vision tasks.

## 2    Characterizing Improvements on ImageNet

Since the breakthrough of AlexNet [30] on ImageNet [45], a wide variety of improvements have been proposed to further advance image recognition performance. These improvements broadly arise along four orthogonal axes: *(a) architecture, (b) training/regularization methodology, (c) scaling strategy and (d) using additional training data*.

**(a) Architecture.**    The works that perhaps receive the most attention are novel architectures. Notable proposals since AlexNet include VGG [49], ResNet [13], Inception [52, 53], and ResNeXt [58]. Automated search strategies for designing architectures have further pushed the state-of-the-art [67, 41, 55]. There have also been efforts in going beyond standard ConvNets for image classification, by adapting self-attention [56] to the visual domain  [2, 40, 20, 47, 8, 1].

**(b) Training and Regularization Methods.**    ImageNet progress has simultaneously been boosted by innovations in training (e.g. improved learning rate schedules [34, 12]) and regularization methods, such as dropout [50], label smoothing [53], stochastic depth [22], dropblock [11] and data augmentation [61, 59, 6, 7]. Regularization methods have become especially useful to prevent overfitting when training ever-increasingly larger models [23] on limited data (e.g. 1.2M ImageNet images).

**(c) Scaling Strategies.**    Increasing the model dimensions (width, depth and resolution) has been another successful axis to improve quality [44, 17]. ResNet architectures are typically scaled up by adding layers (depth): ResNets-18 to ResNet-200 and beyond [14, 62]. Wide ResNets [60] and MobileNets [19] instead scale the width. Increasing image resolutions consistently improves performance: EfficientNet uses 600 image resolutions [55] while both ResNeSt [62] and TResNet [43] use 400+ image resolutions for their largest model. In an attempt to systematize these heuristics, EfficientNet proposed the compound scaling rule, which jointly scales network depth, width and image resolution using a constant scaling factor. However, Section 7.1 shows this scaling strategy is sub-optimal for not only ResNets, but EfficientNets as well.

**(d) Additional Training Data.**    Finally, ImageNet accuracy is commonly improved by training on additional sources of data (either labeled, weakly labeled, or unlabeled). Pre-training on large-scale datasets [51, 35, 27] has significantly pushed the state-of-the-art, with ViT [8] and NFNets [3] recently achieving 88.6% and 89.2% ImageNet accuracy respectively. Using pseudo-labels on additional unlabeled images [57, 37] in a semi-supervised learning fashion has also been a fruitful avenue for improving accuracy. We present semi-supervised learning results in Section 7.2.

## 3    Related Work

Improved training methods combined with architectural changes to ResNets have routinely yielded competitive ImageNet performance [15, 31, 43, 62, 1, 3]. [15] achieved 79.2% top-1 ImageNet accuracy (a +3% improvement over their ResNet-50 baseline) by modifying the stem and downsampling block and using label smoothing and mixup. [31] further improved the ResNet-50 model with additional architectural modifications such as Squeeze-and-Excitation [21], selective kernel [32], and anti-alias downsampling [63], while also using label smoothing, mixup, and dropblock to achieve 81.4% accuracy. [43, 62] incorporate several architectural modifications to the ResNet architectures

along with improved training methodologies to outperform EfficientNet models on the speed-accuracy Pareto curve on GPUs. Many prior works do remark the importance of improved training and regularization methods. However experiments are still largely concerned with architectural changes and the simultaneous introduction of improved training techniques can make it hard to identify where the gains come from[1].

Additionally, due to the ever-increasing performance of machine learning accelerators, newer architectures are routinely pushed to much larger scales than the original ResNets [13]. As a result, works that propose novel architectures do not (cannot) compare against properly trained and scaled ResNets (since such a baseline did not exist), making it challenging to evaluate the significance of the proposed architectural changes compared to simple ResNets.

Lastly, prior work often puts little emphasis on studying scaling strategies or advocates for scaling strategies which we find to be sub-optimal. For example, the largest models in EfficientNet [55], TResNet [43] and ResNeSt [62] use 600, 448 and 416 image sizes respectively, which our scaling analysis reveals to be excessively large. RegNet [39] advocates for width scaling, which we find only works well when overfitting does not occur (e.g. 10 epochs).

In contrast to other works, we only consider lightweight architectural changes (that are widely used since 2018) and keep the architecture fixed. Instead, we focus exclusively on training and scaling strategies to build a Pareto curve of models. Perhaps surprisingly, we find that doing so suffices to outperform models that were introduced after ResNets: our improved training and scaling methods lead to ResNets that are significantly faster than EfficientNets on TPUs on GPUs (see Section 7.1). We note that our scaling improvements are sometimes orthogonal to the architectural innovations introduced in prior works in which case we expect them to be additive.

## 4   Methodology

**Architecture.** Our work studies the ResNet architecture, with two widely used architecture changes, the ResNet-D [15] modification and Squeeze-and-Excitation (SE) in all bottleneck blocks [21]. These architectural changes are used in used many architectures, including TResNet, ResNeSt and EfficientNets. The exact details of our architecture can be found in Appendix E. In our experiments, we sometimes use the original ResNet implementation without SE (referred to as ResNet) to compare different training methods. Clear denotations are made in table captions when this is the case.

**Regularization and Data Augmentation.** We apply *weight decay*, *label smoothing*, *dropout* and *stochastic depth* for regularization. Dropout [50] is a common technique used in computer vision and we apply it to the output after the global average pooling occurs in the final layer. Stochastic depth [22] drops out each layer in the network (that has residual connections around it) with a specified probability that is a function of the layer depth. We use RandAugment [7] data augmentation as an additional regularizer. RandAugment applies a sequence of random image transformations (e.g. translate, shear, color distortions) to each image independently during training. Our training method closely matches that of EfficientNet, where we train for 350 epochs, but with a few small differences (e.g. we use Momentum with cosine learning rate schedule as opposed to RMSProp with exponential decay). See Appendix D for details.

**Hyperparameter Tuning.** To select the hyperparameters for the various regularization and training methods, we use a held-out validation set comprising 2% of the ImageNet training set (20 shards out of 1024). This is referred to as the `minival-set` and the original ImageNet validation set (the one reported in most prior works) is referred to as `validation-set`. Unless specified otherwise, results are reported on the `validation-set`. The hyperparameters of all ResNet-RS models are in Table 8 in the Appendix C.

---

[1]A notable exception is RegNet [39] which purposely makes no use of improved training techniques and shows improvements over worsened EfficientNet baselines, but does not demonstrate ImageNet accuracies above (a rather low) 81%. While this approach facilitates fair comparisons with prior work, it is unclear whether improvements are sustained at larger scales with improved training setups. For example, our scaling analysis shows that the scaling strategy advocated by RegNet does not generalize to training regimes where overfitting can occur.

# 5 Improved Training Methods

## 5.1 Additive Study of Improvements

We present an additive study of training, regularization methods and architectural changes in Table 1 (left). The baseline ResNet-200 gets 79.0% top-1 accuracy. We improve its performance to 82.2% (+3.2%) through *improved training methods alone* without any architectural changes. Adding two common and simple architectural changes (Squeeze-and-Excitation and ResNet-D) further boosts the performance to 83.4%. Training methods alone cause $3/4$ of the total improvement, which demonstrates their critical impact on ImageNet performance.

| Improvements | Top-1 | Δ |
|---|---|---|
| ResNet-200 - 256x256 | 79.0 | — |
| + Cosine LR Decay | 79.3 | **+0.3** |
| + Increase training epochs | 78.8 [†] | -0.5 |
| + EMA of weights | 79.1 | **+0.3** |
| + Label Smoothing | 80.4 | **+1.3** |
| + Stochastic Depth | 80.6 | **+0.2** |
| + RandAugment | 81.0 | **+0.4** |
| + Dropout on FC | 80.7 [‡] | -0.3 |
| + Decrease weight decay | 82.2 | **+1.5** |
| + Squeeze-and-Excitation | 82.9 | **+0.7** |
| + ResNet-D | 83.4 | **+0.5** |

| Model | Regularization | Weight Decay | |
|---|---|---|---|
| | | 1e-4 | 4e-5 |
| ResNet-50 | None | 79.7 | 78.7 (-1.0) |
| ResNet-50 | RA-LS | 82.4 | 82.3 (-0.1) |
| ResNet-50 | RA-LS-DO | 82.2 | 82.7 (+0.5) |
| ResNet-200 | None | 82.5 | 81.7 (-0.8) |
| ResNet-200 | RA-LS | 85.2 | 84.9 (-0.3) |
| ResNet-200 | RA-LS-SD-DO | 85.3 | 85.5 (+0.2) |

Table 1: **(Left) Additive study of training, regularization and architecture improvements.** The baseline ResNet-200 is trained at resolution 256×256 for the standard 90 epochs using a stepwise learning rate decay schedule. All numbers are reported on the ImageNet `validation-set` and averaged over 2 runs. [†] Increasing training duration to 350 epochs only becomes useful once the regularization methods are used, otherwise the accuracy drops due to over-fitting. [‡] dropout hurts as we have not yet decreased the weight decay. **(Right) Decreasing weight decay improves performance when combining regularization methods** such as dropout (DO), stochastic depth (SD), label smoothing (LS) and RandAugment (RA). Image resolution is 224×224 for ResNet-50 and 256×256 for ResNet-200. All numbers are reported on the ImageNet `minival-set` from an average of two runs.

## 5.2 Importance of decreasing weight decay when combining regularization methods

Table 1 (right) highlights the importance of changing weight decay when combining regularization methods together. When applying RandAugment and label smoothing, there is no need to change the default weight decay of 1e-4. But when we further add dropout and/or stochastic depth, the performance can decrease unless we further decrease the weight decay. The intuition is that since weight decay acts as a regularizer, its value must be decreased in order to not overly regularize the model when combining many techniques. Furthermore, [65] presents evidence that the addition of data augmentation shrinks the L2 norm of the weights, which renders some of the effects of weight decay redundant. Other works use smaller weight decay values, but do not point out the significance of the effect when using more regularization [54, 55].

# 6 Improved Scaling Strategies

The prior section demonstrates the significant impact of training methodology and we now show the scaling strategy is similarly important. In order to establish scaling trends, we perform an extensive search on ImageNet over width multipliers in `[0.25, 0.5, 1.0, 1.5, 2.0]`, depths of `[26, 50, 101, 200, 300, 350, 400]` and resolutions of `[128, 160, 224, 320, 448]`. We train these architectures for 350 epochs, mimicking the training setup of state-of-the-art ImageNet models, and increase regularization with model size in an effort to limit overfitting. See Appendix F for regularization and model hyperparameters.

**FLOPs do not accurately predict performance in the bounded data regime.** Prior works on scaling laws observe a power law between error and FLOPs in *unbounded data regimes* [25, 16]. In order to test whether this also holds in our scenario, we plot ImageNet error against FLOPs for all scaling configurations in Figure 2.

For the smaller models, we observe an overall power law trend between error and FLOPs, with minor dependency on the scaling configuration (i.e. depth, width and image resolution). However, the trend breaks for larger model sizes and we observe a large variation in ImageNet performance for a fixed amount of FLOPs, especially in the higher FLOP regime. Therefore the exact scaling configuration (i.e. depth, width and image resolution) can have a big impact on performance even when controlling for the same amount of FLOPs.

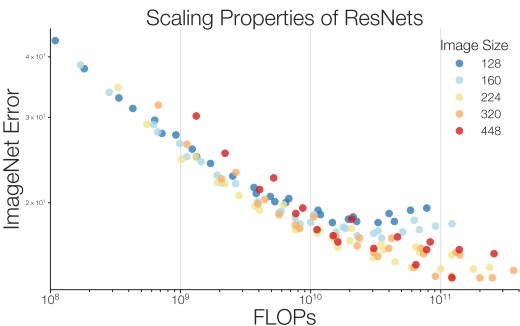

Figure 2: **Scaling properties of ResNets across varying model scales.** Error approximately scales as a power law with FLOPs (linear fit on the log-log curve) in the lower FLOPs regime but the trend breaks for larger FLOPs. We observe diminishing returns of scaling the image resolutions beyond 320×320, which motivates the slow image resolution scaling (Strategy #2). All results are on the ImageNet `minival-set`.

**The best performing scaling strategy depends on the training regime.** We next look directly at latencies[2] on the hardware of interest to identify scaling strategies that improve the speed-accuracy Pareto curve. Figure 3 presents accuracies and latencies of models scaled with either width or depth across four image resolutions and three different training regimes (10, 100 and 350 epochs). We observe that the best performing scaling strategy, especially whether to scale depth and/or width, highly depends on the training regime.

## 6.1 Strategy #1 - Depth Scaling in Regimes Where Overfitting Can Occur

**Depth scaling outperforms width scaling for longer epoch regimes.** In the 350 epochs setup (Figure 3 - right), we observe depth scaling to significantly outperform width scaling across all image resolutions. Scaling the width is subject to overfitting and sometimes hurts performance even with increased regularization. We hypothesize that this is due to the larger increase in parameters when scaling the width. The ResNet architecture maintains constant FLOPs across all block groups and multiplies the number of parameters by 4× every block group. Scaling the depth, especially in the earlier layers, therefore introduces fewer parameters compared to scaling the width.

**Width scaling outperforms depth scaling for shorter epoch regimes.** In contrast, width scaling is better when only training for 10 epochs (Figure 3 - left). For 100 epochs (Figure 3 - middle), the best performing scaling strategy varies between depth scaling and width scaling, depending on the image resolution. The dependency of the scaling strategy on the training regime reveals a pitfall of extrapolating scaling rules. We point out that prior works also choose to scale the width when training for a small number of epochs on large-scale datasets (e.g. ∼40 epochs on 300M images), consistent with our experimental findings that scaling the width is preferable in shorter epoch regimes. In particular, [27] train a ResNet-152 with 4x filter multiplier while [3] scales the width with ∼1.5x filter multiplier.

## 6.2 Strategy #2 - Slow Image Resolution Scaling

In Figure 2, we also observe that larger image resolutions yield diminishing returns. We therefore propose to increase the image resolution more gradually than previous works. This contrasts with the compound scaling rule proposed by EfficientNet which leads to very large images (e.g. 600 for EfficientNet-B7, 800 for EfficientNet-L2 [57]). Other works such as ResNeSt [62] and TResNet [43]) scale the image resolution up to 400+. Our experiments indicate that slower image scaling improves not only ResNet architectures, but also EfficientNets on a speed-accuracy basis (Section 7.1).

---

[2]FLOPs is not a good indicator of latency on modern hardware. See Section 7.1 for a more detailed discussion.

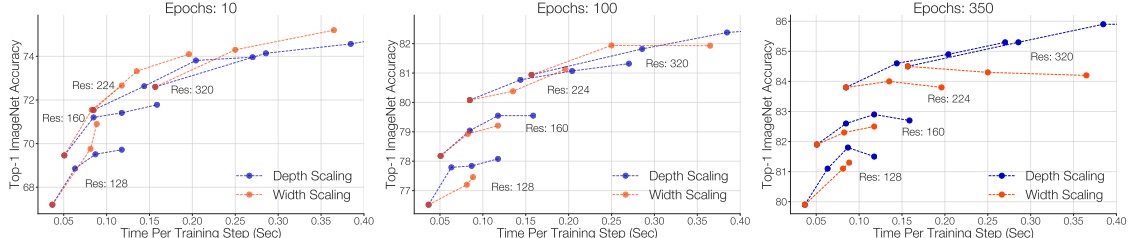

Figure 3: **Scaling of ResNets across depth, width, image resolution and training epochs**. We compare depth scaling and width scaling across four different image resolutions [128,160,224,320] when training models for 10, 100 or 350 epochs. We find that *the best performing scaling strategy depends on the training regime*, which reveals the pitfall of extrapolating scaling rules from small scale regimes. **(Left) 10 Epoch Regime**: width scaling is the best strategy for the speed-accuracy Pareto curve. **(Middle) 100 Epoch Regime**: depth scaling is sometimes outperformed by width scaling. **(Right) 350 Epoch Regime**: depth scaling consistently outperforms width scaling by a large margin. Overfitting remains an issue even when using regularization methods. **Model Details:** All models start from a depth of 101 and are increased through [101,200,300,400]. All model widths start with a multiplier of 1.0x and are increased through [1.0,1.5,2.0]. For all models, we tune regularization in an effort to limit overfitting (see Appendix F). Accuracies are reported on the ImageNet minival-set and training times are measured on TPUs.

## 6.3 Designing Scaling Strategies

Our scaling analysis surfaces two common pitfalls in prior research on scaling strategies.

**Pitfall #1: Extrapolating scaling strategies from small-scale regimes.** Scaling strategies found in small scale regimes (e.g. on small models or with few training epochs) can fail to generalize to larger models or longer training iterations. The dependencies between the best performing scaling strategy and the training regime are missed by prior works which extrapolate scaling rules from either small models [55] or shorter training epochs [39]. We therefore do not recommend generating scaling rules exclusively in a small scale regime because these rules can break down.

**Pitfall #2: Extrapolating scaling strategies from a single and potentially sub-optimal initial architecture.** Beginning from a sub-optimal initial architecture can skew the scaling results. For example, the compound scaling rule derived from a small grid search around EfficientNet-B0, which was obtained by architecture search using a fixed FLOPs budget and a specific *image resolution*. However, since this image resolution can be sub-optimal for that FLOPs budget, the resulting scaling strategy can be sub-optimal. In contrast, our work designs scaling strategies by training models across a variety of widths, depths and image resolutions.

**Summary of Improved Scaling Strategies.** For image classification, the scaling strategies are summarized as **(1)** scale the depth in regimes where overfitting can occur (scaling the width is preferable otherwise) and **(2)** slow image resolution scaling. Experiments indicate that applying these scaling strategies to ResNets (ResNet-RS) and EfficientNets (EfficientNet-RS) leads to significant speed-ups over EfficientNets. We note that similar scaling strategies are also employed in recent works that obtain large speed-ups over EfficientNets such as LambdaResNets [1] and NFNets [3]. For a new task, we recommend running a *small subset* of models across different scales, for the full training epochs, to gain intuition on which dimensions are the most useful across model scales. While this approach may appear more costly, we point out that the cost is offset by not searching for the architecture.

## 7 Experiments with Improved Training and Scaling Strategies

### 7.1 ResNet-RS on a Speed-Accuracy Basis

Using the improved training and scaling strategies, we design *ResNet-RS*, a family of re-scaled ResNets across a wide range of model scales (see Appendix C and E for experimental and architectural details). Figure 4 and Table 2 compare EfficientNets against ResNet-RS on a speed-accuracy Pareto curve. We find that ResNet-RS match EfficientNets' performance while being **1.7x - 2.7x** faster on TPUs (**2.1x - 3.3x** faster on GPUs). We point that these speed-ups are superior to those obtained by TResNest and ResNeSt[3], suggesting that ResNet-RS also outperform TResNet and ResNeSt.

---

[3]TResNet and ResNeSt report ∼1.3 - 2.0x speed-ups over EfficientNet on a GPU V100.

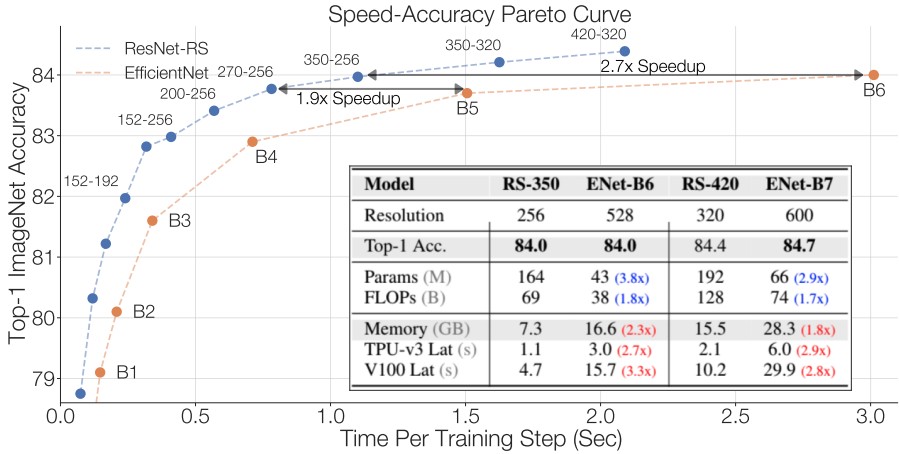

Figure 4: **Speed-Accuracy Pareto curve comparing ResNets-RS to EfficientNet.** ResNet-RS (annotated with depth - image resolution) are **1.7x - 2.7x** faster than the popular EfficientNets when closely matching their training setup. Although ResNet-RS has more parameters and FLOPs, the model employs less memory and runs faster on TPUs and GPUs. See Appendix C and I for more results and profiling details.

| Model | Image Resolution | Params (M) | FLOPs (B) | V100 Latency (s) | TPUv3 Latency (ms) | Top-1 |
|---|---|---|---|---|---|---|
| EfficientNet-B0 | 224 | 5.3 | 0.8 | 0.47 | 90 | 77.1 |
| EfficientNet-B1 | 240 | 7.8 | 1.4 | 0.82 | 150 | 79.1 |
| ResNet-RS-50 | 160 | 36 | 4.6 | 0.31 | 70 | 78.8 |
| EfficientNet-B2 | 260 | 9.2 | 2.0 | 1.03 | 210 | 80.1 |
| ResNet-RS-101 | 160 | 64 | 8.4 | 0.48 (**2.1×**) | 120 (**1.8×**) | 80.3 |
| EfficientNet-B3 | 300 | 12 | 3.6 | 1.76 | 340 | 81.6 |
| ResNet-RS-101 | 192 | 64 | 12 | 0.70 | 170 | 81.2 |
| ResNet-RS-152 | 192 | 87 | 18 | 0.99 | 240 | 82.0 |
| EfficientNet-B4 | 380 | 19 | 8.4 | 4.0 | 710 | 82.9 |
| ResNet-RS-152 | 224 | 87 | 24 | 1.48 (**2.7×**) | 320 (**2.2×**) | 82.8 |
| ResNet-RS-152 | 256 | 87 | 31 | 1.76 (**2.3×**) | 410 (**1.7×**) | 83.0 |
| EfficientNet-B5 | 456 | 30 | 20 | 8.16 | 1510 | 83.7 |
| ResNet-RS-200 | 256 | 93 | 40 | 2.86 | 570 | 83.4 |
| ResNet-RS-270 | 256 | 130 | 54 | 3.76 (**2.2×**) | 780 (**1.9×**) | 83.8 |
| EfficientNet-B6 | 528 | 43 | 38 | 15.7 | 3010 | 84.0 |
| ResNet-RS-350 | 256 | 164 | 69 | 4.72 (**3.3×**) | 1100 (**2.7×**) | 84.0 |
| EfficientNet-B7 | 600 | 66 | 74 | 29.9 | 6020 | 84.7 |
| ResNet-RS-350 | 320 | 164 | 107 | 8.48 | 1630 | 84.2 |
| ResNet-RS-420 | 320 | 192 | 128 | 10.16 | 2090 | 84.4 |

Table 2: **Details of ResNet-RS models in Pareto curve.** See Table 8 for hyperparameters and Section I for profiling details.

This large speed-up over EfficientNet may be non-intuitive since EfficientNets have significantly reduced parameters and FLOPs compared to ResNets. We next discuss why a model with fewer parameters and fewer FLOPs (EfficientNet) is slower and more memory-intensive during training.

**FLOPs vs Latency.** While FLOPs provide a hardware-agnostic metric for assessing computational demand, they may not be indicative of actual latency times for training and inference [19, 18, 39]. In custom hardware architectures (e.g. TPUs and GPUs), FLOPs are an especially poor proxy because operations are often bounded by memory access costs and have different levels of optimization on modern matrix multiplication units [24]. The inverted bottlenecks [46] used in EfficientNets employ depthwise convolutions with large activations and have a *small compute to memory ratio* (operational intensity) compared to the ResNet's bottleneck blocks which employ dense convolutions on smaller activations. This makes EfficientNets less efficient on modern accelerators compared to ResNets.

Figure 4 (table on the right) illustrates this point: a ResNet-RS model with **1.8x** more FLOPs than EfficientNet-B6 is **2.7x** faster on a TPUv3 hardware accelerator.

**Parameters vs Memory.** Parameter count does not necessarily dictate memory consumption during *training* because memory is often dominated by the size of the *activations*[4]. The large activations used in EfficientNets also cause larger memory consumption, which is exacerbated by the use of large image resolutions, compared to our re-scaled ResNets. A ResNet-RS model with **3.8x** more parameters than EfficientNet-B6 consumes **2.3x** less memory for a similar ImageNet accuracy (Table in Figure 4). We emphasize that both memory consumption and latency are tightly coupled to the software and hardware stack (TensorFlow on TPUv3) due to compiler optimizations such as operation layout assignments and memory padding.

**Improving scaling of EfficientNets** The scaling analysis from Section 6 reveals that scaling the image resolution results in diminishing returns. This suggests that the compound scaling rule advocated in EfficientNet which jointly increases model depth, width and resolution at a constant rate is sub-optimal. To test this hypothesis, we apply the slow image resolution scaling strategy (Strategy #2) to EfficientNets and train several versions with reduced image resolutions, without changing the width or depth. Figure 5 (Appendix) demonstrates a marked improvement of the re-scaled EfficientNets (EfficientNet-RS) on the speed-accuracy Pareto curve over the original EfficientNets.

## 7.2 Semi-Supervised Learning with ResNet-RS

We next measure how ResNet-RS performs as we scale to larger datasets in a large scale semi-supervised learning setup. We train ResNets-RS on the combination of 1.3M labeled ImageNet images and 130M *pseudo-labeled* images, in a similar fashion to Noisy Student [57]. We use the same dataset of 130M images pseudo-labeled as Noisy Student, where the pseudo labels are generated from an EfficientNet-L2 model with 88.4% ImageNet accuracy.

Models are jointly trained on both the labeled and pseudo-labeled data and training hyperparameters are kept the same. Table 3 reveals that ResNet-RS models are very strong in the semi-supervised learning setup as well, achieving a strong 86.2% top-1 ImageNet accuracy while being **4.7x** faster on TPU (**5.5x** on GPU) than the corresponding EfficientNet model.

| Model | V100 (s) | TPUv3 (ms) | Top-1 |
|---|---|---|---|
| EfficientNet-B5 | 8.16 | 1510 | 86.1 |
| ResNet-RS-152 | **1.48 (5.5x)** | **320 (4.7x)** | **86.2** |

Table 3: **ResNet-RS are efficient semi-supervised learners.** ResNet-RS-152 with image resolution 224 is **4.7x** faster on TPU (**5.5x** on GPU) than EfficientNet-B5 Noisy Student for a similar ImageNet accuracy.

## 7.3 Transfer Learning to Downstream Tasks with ResNet-RS

We now investigate whether the improved supervised training strategies yield better representations for transfer learning and compare them with self-supervised learning algorithms. Recent self-supervised learning algorithms claim to surpass the transfer learning performance of *supervised learning* and create more universal representations [4, 5]. Self-supervised algorithms, however, make several changes to the training methods (e.g training for more epochs, data augmentation) making comparisons to supervised learning difficult.

**Fairly comparing supervised learning and self-supervised learning.** In an effort to closely match SimCLR's training setup and provide fair comparisons, we restrict the RS training strategies to a subset of its original methods. Specifically, we train for for 400 epochs with cosine learning rate decay, data augmentation (RandAugment), label smoothing, dropout and decreased weight decay but do not use stochastic depth or exponential moving average (EMA) of the weights. We choose this subset to closely match the training setup of SimCLR: longer training (800 epochs) with cosine learning rate decay, a tailored data augmentation strategy, a tuned temperature parameter in the contrastive loss and a tuned weight decay.

Table 4 compares the transfer performance of supervised learning with or without improved training strategies (respectively denoted RS and Supervised) against SimCLR/SimCLRv2 [4, 5] on five downstream tasks: CIFAR-100 Classification [29], Pascal Detection & Segmentation [9], ADE Segmentation [64] and NYU Depth [48]. Our experiments demonstrate that the improved training

---

[4]Activations are typically stored during training as they are used in backpropagation. At inference, activations can be discarded and parameter count is a better proxy for actual memory consumption.

| Model | Training Method | Epochs | CIFAR-100 Accuracy | Pascal Detection | Pascal Segmentation | ADE Segmentation | NYU Depth |
|---|---|---|---|---|---|---|---|
| ResNet-152 | Supervised | 90 | 85.5 | 80.0 | 70.0 | 40.2 | 81.2 |
| ResNet-152 | SimCLR | 800 | 87.1 | **83.3** | 72.2 | 41.0 | 83.5 |
| ResNet-152 | SimCLRv2 | 800 | 84.7 | 79.1 | 73.1 | 41.1 | **84.7** |
| ResNet-152 | RS | 400 | **88.1** | 82.2 | **78.2** | **42.2** | 83.4 |

Table 4: **Representations from supervised learning with improved training strategies rival or outperform representations from state-of-the-art self-supervised learning algorithms.** Comparison of supervised training methods (supervised, RS) and self-supervised methods (SimCLR, SimCLRv2) on a variety of downstream tasks. The improved training strategies (RS) greatly outperforms the baseline supervised training, which highlights the importance of using improved supervised training techniques when comparing to self-supervised learning algorithms. All models employ the *vanilla* ResNet architecture and are pre-trained on ImageNet.

strategies significantly improve transfer performance, in line with works that observe that higher ImageNet accuracy strongly correlates with improved transfer learning performance [28]. Furthermore, we find that the improved supervised representations (RS) rival or outperform SimCLR/SimCLRv2, even when restricted to a smaller subset. These results challenge the notion that self-supervised algorithms lead to more universal representations than supervised learning when labels are available.

### 7.4 Revised 3D ResNet for Video Classification

We conclude by applying the training strategies to the Kinetics-400 video classification task [26], using a 3D ResNet as the baseline architecture [38]. Table 5 presents an additive study of the RS training recipe and architectural improvements. The training strategies extend to video classification, yielding a combined improvement from 73.4% to 77.4% (+4.0%). The ResNet-D and Squeeze-and-Excitation architectural changes further improve the performance to 78.2% (+0.8%). Similarly to our study on image classification (Table 1), we find that most of the improvement can be obtained without architectural changes.

| Improvements | Top-1 | Δ |
|---|---|---|
| 3D ResNet-50 | 73.4 | – |
| + Dropout on FC | 74.4 | **+1.0** |
| + Label smoothing | 74.9 | **+0.5** |
| + Stochastic depth | 76.1 | **+1.2** |
| + EMA of weights | 76.1 | – |
| + Decrease weight decay | 76.3 | **+0.2** |
| + Increase training epochs | 76.4 | **+0.1** |
| + Scale jittering | 77.4 | **+1.0** |
| + Squeeze-and-Excitation | 77.9 | **+0.5** |
| + ResNet-D | 78.2 | **+0.3** |

Table 5: **Additive study of regularization , training and architecture improvements with 3D-ResNet on video classification.**

## 8 Conclusion

By updating the de facto vision baseline with modern training methods and an improved scaling strategy, we have revealed the remarkable durability of the ResNet architecture. Simple architectures set strong baselines for state-of-the-art methods: the accuracy gains that motivate complicated architectural changes may be surpassed with thoughtful scaling and training strategies. Our work suggests that the field has myopically overemphasized architectural innovations at the expense of experimental diligence, and we hope it encourages further scrutiny in maintaining consistent methodology for both proposed innovations and baselines alike. We do not foresee any negative societal impact of our work. We include further discussion in the Appendix B.

**Acknowledgements.** We would like to thank Ashish Vaswani, Prajit Ramachandran, Ting Chen, Thang Luong, Hanxiao Liu, Gabriel Bender, Quoc Le, Neil Houlsby, Mingxing Tan, Andrew Howard, Raphael Gontijo Lopes, Andy Brock and David Berthelot for helpful feedback on this work; Jing Li, Pengchong Jin, Yeqing Li and Yin Cui for the support on open-sourcing and infrastructure.

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
