# A  Author Contributions

**IB, BZ:** led the research, designed and ran the scaling experiments, designed and experimented with the training strategies. **JS, TL, EC, AS, WF, XD:** advised the research, proposed experiments and helped with the writing. **AS, IB, BZ:** ran preliminary experiments using label smoothing, longer training and RandAugment. **IB:** demonstrated ResNets outperforming EfficientNets across all scales, designed the scaling strategies and the Pareto curve of models, designed/ran (semi-)supervised learning experiments. **BZ:** ran the regularization studies. **WF, BZ, IB:** did a majority of the writing. **BZ, EC:** analyzed scaling experiments and generated the scaling plots. **XD:** proposed, designed and ran the 3D video classification experiments, lead the open-sourcing. **AS:** proposed lowering the weight decay for better performance and ran preliminary experiments comparing SimCLR to supervised learning. **TL:** designed and ran the transfer learning experiments comparing to self-supervised learning.

# B  Discussion

**Why is it important to tease apart improvements coming from training methods vs architectures?**    Training methods can be more task-specific than architectures (e.g. data augmentation is more helpful on small datasets). Therefore, improvements coming from training methods do not necessarily generalize as well as architectural improvements. Packaging newly proposed architectures together with training improvements makes accurate comparisons between architectures difficult. The large improvements coming from training strategies, when not being controlled for, can overshadow architectural differences.

**How should one compare different architectures?**    Since training methods and scale typically improve performance [31, 25], it is critical to control for both aspects when comparing different architectures. Controlling for scale can be achieved through different metrics. While many works report parameters and FLOPs, we argue that latencies and memory consumption are generally more relevant [39]. Our experimental results (Section 7.1) re-emphasize that FLOPs and parameters are not representative of latency or memory consumption [39, 36].

**Do the improved training strategies transfer across tasks?**    The answer depends on the domain and dataset sizes available. Many of the training and regularization methods studied here are not used in large-scale pre-training (e.g. 300M images) [27, 8]. Data augmentation is useful for small datasets or when training for many epochs, but the specifics of the augmentation method can be task-dependent (e.g. scale jittering instead of RandAugment in Table 5).

**Do the scaling strategies transfer across tasks?**    The best performing scaling strategy depends on the training regime and whether overfitting is an issue, as discussed in Section 6. When training for 350 epochs on ImageNet, we find scaling the depth to work well, whereas scaling the width is preferable when training for few epochs (e.g. 10 epochs). This is consistent with works employing width scaling when training for few epochs on large-scale datasets [27]. We are unsure how our scaling strategies apply in tasks that require larger image resolutions (e.g. detection and segmentation) and leave this to future work.

**Are architectural changes useful?**    Yes, but training methods and scaling strategies can have even larger impacts. Simplicity often wins, especially given the non-trivial performance issues arising on custom hardware. Architecture changes that decrease speed and increase complexity may be surpassed by scaling up faster and simpler architectures that are optimized on available hardware (e.g convolutions instead of depthwise convolutions for GPUs/TPUs). We envision that future successful architectures will emerge by co-design with hardware, particularly in resource-tight regimes like mobile phones [18].

**How should one allocate a computational budget to produce the best vision models?**    We recommend beginning with a simple architecture that is efficient on available hardware (e.g. ResNets on GPU/TPU) and training several models, to convergence, with different image resolutions, widths and depths to construct a Pareto curve. Note that this strategy is distinct from [55] which instead allocate a large portion of the compute budget for identifying an optimal initial architecture to scale. They then do a small grid search to find the compound scaling coefficients used across all model scales. RegNet [39] does most of their studies when training for only 10 epochs.

## C  Additional Experimental Results

**Scaling Strategies Improve EfficientNet.**   We apply the slow image resolution scaling strategy (Strategy #2) to EfficientNets and train several versions with reduced image resolutions, without changingthe width or depth. Figure 5 demonstrates a marked improvement of the re-scaled Efficient-Nets (EfficientNet-RS) on the speed-accuracy Pareto curve over the original EfficientNets.

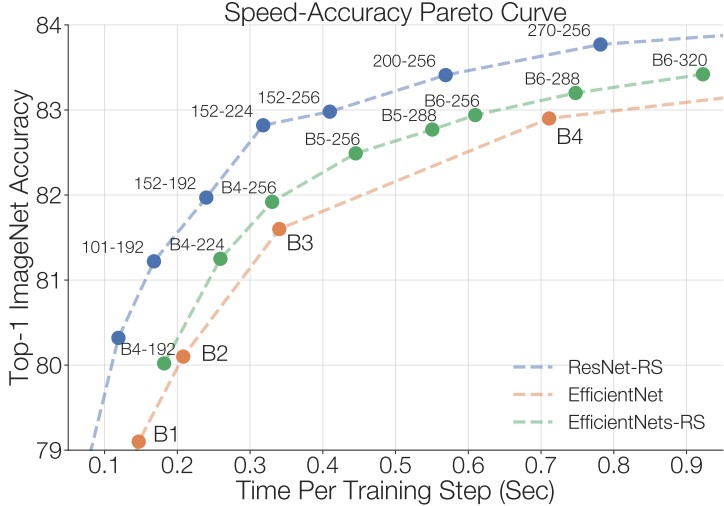

Figure 5: **Speed-Accuracy Pareto curve comparing ResNets-RS and EfficientNet-RS to EfficientNet.** Scaling EfficientNets using the slow image resolution scaling strategy (instead of the original compound scaling rule) improves the Pareto efficiency of EfficientNets. Note that ResNet-RS still outperforms EfficientNet-RS. This figure is a zoomed in version of Figure 4 with EfficientNet-RS added. Models are annotated with (model depth - image resolution), so 152-192 corresponds to ResNet-RS-152 with image resolution 192×192. **EfficientNet hyperparameters:** The RandAugment magnitude is set to 10 for image resolution 224 or smaller, 20 for image resolution larger than 320 and 15 otherwise. All other hyperparameters are kept the same as per the original EfficientNets.

**ImageNet test set results.**   We present top-1 accuracies on the ImageNet `test-set` for two ResNet-RS models in Table 6. We observe no sign of overfitting.

| Model | Image Resolution | top-1 Val | top-1 Test |
|---|---|---|---|
| ResNet-RS-152 | 224 | 82.8 | 82.7 |
| ResNet-RS-270 | 256 | 83.8 | 83.7 |

Table 6: **ImageNet accuracies on the validation and test splits.**

## D  ResNet-RS Training and Regularization Methods

Our training methods aim to closely match that of EfficientNet [55], but with a few small differences listed below. **(1)** We use the cosine learning rate schedule [34] instead of an exponential decay for simplicity (no additional hyperparameters). **(2)** We use RandAugment [7] in all models, whereas EfficientNets were originally trained with AutoAugment [6]. We reran EfficientNets B0-B4 with RandAugment and found it offered no performance improvement and report EfficientNet B5 and B7 with the RandAugment results from [7][5]. **(3)** We use the Momentum optimizer instead of RMSProp for simplicity. See Table D for a comparison between our training setup and EfficientNet.

---

[5]This makes our comparison to EfficientNet-B6 more nuanced as the B6 performance most likely could be improved by 0.1-0.3% top-1 if ran with RandAugment (based on improvements obtained from B5 and B7).

|  | ResNet (2015) | ResNet-RS (2021) | EfficientNets (2019) |
|---|---|---|---|
| Epochs Trained | 90 | 350 | 350 |
| LR Decay Schedule | Stepwise | Cosine | Exponential Decay |
| Optimizer | Momentum | Momentum | RMSProp |
| EMA of Weights |  | ✓ | ✓ |
| Label Smoothing |  | ✓ | ✓ |
| Stochastic Depth |  | ✓ | ✓ |
| RandAugment |  | ✓ | ✓ |
| Dropout on FC |  | ✓ | ✓ |
| Smaller Weight Decay |  | ✓ | ✓ |
| Squeeze-Excitation |  | ✓ | ✓ |
| Stem Modifications |  | ✓ | ✓ |

Table 7: **Comparing training method between ResNet, ResNet-RS and EfficientNet.** ResNet (2015) refers to the ResNet originally trained in [13].

**Hyperparameters**  Table 8 presents the training and regularization hyperparameters used for training ResNet-RS models. We increase regularization as with model scale. Note that we have less hyperparameter setups compared to EfficientNets [55]. We perform early stopping on the `minival-set` set for the two largest models from Table 2 (ResNet-RS-350 at resolution 320 and ResNet-RS-420 at resolution 320).

| Model | Depth | Image Resolution | RandAugment Magnitude | Stochastic Depth Rate | Dropout Rate |
|---|---|---|---|---|---|
| ResNet-RS | 50 | $160 \times 160$ | 10 | 0.0 | 0.25 |
| ResNet-RS | 101 | $160 \times 160$ | 10 | 0.0 | 0.25 |
| ResNet-RS | 101 | $192 \times 192$ | 15 | 0.0 | 0.25 |
| ResNet-RS | 152 | $192 \times 192$ | 15 | 0.0 | 0.25 |
| ResNet-RS | 152 | $224 \times 224$ | 15 | 0.0 | 0.25 |
| ResNet-RS | 152 | $256 \times 256$ | 15 | 0.0 | 0.25 |
| ResNet-RS | 200 | $256 \times 256$ | 15 | 0.1 | 0.25 |
| ResNet-RS | 270 | $256 \times 256$ | 15 | 0.1 | 0.25 |
| ResNet-RS | 350 | $256 \times 256$ | 15 | 0.1 | 0.25 |
| ResNet-RS | 350 | $320 \times 320$ | 15 | 0.1 | 0.4 |
| ResNet-RS | 420 | $320 \times 320$ | 15 | 0.1 | 0.4 |

Table 8: **Hyperparameters for all ResNet-RS models.** All models train for 350 epochs, use a weight decay of 4e-5, an EMA value of 0.9999 (for both weights and Batch Norm moving averages), 2 layers of RandAugment (with different magnitudes as shown above) and a label smoothing rate of 0.1. The learning rate is warmed up to a maximum value of $0.1/B$, with B the batch size, and decayed to 0 using a cosine schedule [34]. Dropout rate means each activation after the global average pooling layers gets dropped out with probability *dropout rate*.

# E ResNet-RS Architecture Details

We provide more details of the ResNet-RS architectural changes. We reiterate that ResNet-RS is a combination of: improved scaling strategies, improved training methodologies, the ResNet-D modifications [15] and the Squeeze-Excitation module [21].

**ResNet-D** [15] combines the following four adjustments to the original ResNet architecture. First, the $7 \times 7$ convolution in the stem is replaced by three smaller $3 \times 3$ convolutions, as first proposed in Inception-V3 [53]. Second, the stride sizes are switched for the first two convolutions in the residual path of the downsampling blocks. Third, the stride-2 $1 \times 1$ convolution in the skip connection path of the downsampling blocks is replaced by stride-2 $2 \times 2$ average pooling and then a non-strided $1 \times 1$ convolution. Fourth, the stride-2 $3 \times 3$ max pool layer is removed and the downsampling occurs in the first $3 \times 3$ convolution in the next bottleneck block. We diagram these modifications in Figure 6.

**Squeeze-and-Excitation** [21] reweighs channels via cross-channel interactions by average pooling signals from the entire feature map. For all experiments we use a Squeeze-and-Excitation ratio of 0.25 based on preliminary experiments.

Table 9 shows the block layouts for all ResNet depths used throughout our work. ResNet-50 through ResNet-200 use the standard block configurations from [13]. ResNet-270 and onward primarily scale the number of blocks in c3 and c4 and we try to keep their ratio roughly constant. We empirically found that adding blocks in the lower stages limits overfitting as blocks in the lower layers have significantly less parameters, even though all blocks have the same amount of FLOPs. Figure 6 shows the ResNet-D architectural changes used in our ResNet-RS models.

| Model | Depth | Block Configuration |
|-------|-------|---------------------|
| ResNet | 50 | [3-4-6-3] |
| ResNet | 101 | [3-4-23-3] |
| ResNet | 152 | [3-8-36-3] |
| ResNet | 200 | [3-24-36-3] |
| ResNet | 270 | [4-29-53-4] |
| ResNet | 350 | [4-36-72-4] |
| ResNet | 420 | [4-44-87-4] |

Table 9: **Block configurations for all ResNet depths used in the ResNet-RS Pareto Curve.** ResNets of depths 50, 101, 152 and 200 use the standard block allocations from [13]. The different numbers represent the number of blocks in c2, c3, c4 and c5 respectively. Note that our depth scaling mainly scales the blocks in c3 and c4, which limits overfitting (due to the increase in parameters) that can occur when blocks are added to c5.

| Block Group | Output Size | Convolution Layout | |
|-------------|-------------|--------------------|---|
| stem | 112x112 | 3x3, 64, s2
3x3, 64
3x3, 64 | x1 |
| c2 | 56x56 | 1x1, 64
3x3, 64
1x1, 256 | x3 |
| c3 | 28x28 | 1x1, 128
3x3, 128
1x1, 512 | x4 |
| c4 | 14x14 | 1x1, 256
3x3, 256
1x1, 1024 | x23 |
| c5 | 7x7 | 1x1, 512
3x3, 512
1x1, 2048 | x3 |
| | 1x1 | Avg Pool
Dropout
1000-d FC | x1 |

Figure 6: **ResNet-RS Architecture Diagram.** Output Size assumes a 224×224 input image resolution. In the convolutional layout column $x2$ refers to the the first $3 \times 3$ convolution being applied with a stride of 2. The ResNet-RS architecture is a simple combination of Squeeze-and-Excitation and ResNet-D. The × symbol refers to how many times the blocks are repeated in the ResNet-101 architecture. These values change across depths according to the blocks layouts in Table 9.

## F Scaling Analysis Regularization and Model Details

**Regularization for 10 and 100 epochs.** We did not use RandAugment, Dropout, Stochastic Depth or Label Smoothing. Flips and crops were used and a weight decay of 4e-5.

| Filter Scaling | Dropout Rate |
|:---:|:---:|
| 0.25 | 0.0 |
| 0.5 | 0.1 |
| 1.0 | 0.25 |
| 1.5 | 0.6 |
| 2.0 | 0.75 |

Table 10: **Dropout values for filter scaling.** Filter scaling refers to the filter scaling multiplier based on the number of filters in the original ResNet architecture.

**Regularization for 350 epoch models.**    The dropout rates used for various filter multipliers (across all image resolutions and depths) are in Table 10. RandAugment is used with 2 layers and its magnitude is set to 10 for filter multipliers in [0.25, 0.5] or image resolution in [64, 160], 15 for image resolution in [224, 320] and 20 otherwise. We apply stochastic depth with a drop rate of 0.2 for image resolutions 224 and above. We do not apply stochastic depth filter multiplier 0.25 (or images smaller than 224). All models use a label smoothing of 0.1 and a weight decay of 4e-5. These values were set based on the preliminary experiments across various model scales on the ImageNet `minival-set`.

**Block allocation for ResNet-300 and ResNet-400.**  For ResNet 101 and ResNet-200 we use the block allocations decribed in Table 9. For ResNet-300, our block allocation is `[4-36-54-4]` and ResNet-400 is `[6-48-72-6]`.

## G  Fine-Tuning Protocols for Transfer Learning

For fine-tuning we initialize the parameters in the ResNet backbone with a pre-trained model and randomly initialize the rest of the layers. We perform *end-to-end* fine-tuning with an extensive grid search of the combinations of learning rate and training steps to ensure each pre-trained model achieves its best fine-tuning performance. We experiment with different weight decays but do not find it making a big difference and set it to 1e-4. All models are trained with cosine learning rate for simplicity. Below we describe the dataset, evaluation metric, model architecture, and training parameters for each task.

**CIFAR-100:**   We use standard CIFAR-100 train and test sets and report the top-1 accuracy. We resize the image resolution to $256 \times 256$. We replace the classification head in the pre-trained model with a randomly initialized linear layer that predicts 101 classes, including background. We use a batch size of 512 and search the combination of training steps from 5000 to 20000 and learning rates from 0.005 to 0.32. We find the best learning rate for SimCLR (0.16) is much higher than SimCLRv2 (0.01) and the supervised model (0.005). This trend holds for the following tasks.

**PASCAL Segmentation:**   We use PASCAL VOC 2012 train and validation sets and report the mIoU metric. The training images are resampled into $512 \times 512$ with scale jittering [0.5, 2.0] (i.e. randomly resample image between $256 \times 256$ to $1024 \times 1024$ and crop it to $512 \times 512$). We remove the classification head and add randomly initialized FPN [33] layers. We follow the practice in [66] to combine $P_3$ to $P_7$ and upsample it to $P_2$. The segmentation head consists of 3 convolution layers after $P_2$ layer and a linear layer to predict 21 categories including background at each pixel location. We use a batch size of 64 and search the combination of training steps from 5000 to 20000 and learning rates from 0.005 to 0.32.

**PASCAL Detection:**   We use PASCAL VOC 2007+2012 trainval set and VOC 2007 test set and report the $AP_{50}$ with 11 recall points to compute average precision. The training images are resampled into 896 with scale jittering [0.5, 2.0]. We remove the classification head and add randomly initialized FPN [33] layers from $P_3$ to $P_7$. We use Faster R-CNN [42] consisting a region proposal head and a `4conv1fc` Fast R-CNN head. We use a batch size of 32 and search the combination of training steps from 5000 to 20000 and learning rates from 0.005 to 0.32.

**NYU Depth:**   We use NYU depth v2 dataset with 47584 train and 654 validation images. We report the percentage of predicted depth values within 1.25 relative ratio compared to the ground truth. The training images are resampled into 640 with scale jittering [0.5, 2.0]. The model architecture is

identical to segmentation model, except the last linear layer predicts a single depth value per pixel. We use a batch size of 64 and search the combination of training steps from 10000 to 40000 and learning rates from 0.005 to 0.32.

## H   Video Classification Experimental Details

The baseline 3D ResNet-50 was trained for 200 epochs with a cosine learning rate decay. The reported accuracies are averaged over 2 runs.

We follow the training and inference protocols in [38, 10]. We train with a random 224×224 crop or its horizontal flip on the spatial domain and sample a 32-frame clip with temporal stride 2. We use a 1024 batch size, 0.8 learning rate with cosine decay and train for 200 epochs for the baseline. At inference, we use 256×256 crop size for the spatial domain and adopt the 30 views protocol [10].

Starting from the baseline, we apply the following training methods: dropout with a rate of 0.5, 0.1 label smoothing, stochastic depth with 0.2 drop rate, EMA of weights, smaller weight decay (set to 4e-5) and a 350 epoch training schedule. For data augmentation, we use scale jittering [38] as a replacement to RandAugment. We adjust the stochastic depth rate to 0.1 when applying scale jittering to optimize performance. To implement the ResNet-D stem for the 3D ResNet, we use the same kernel configurations for the spatial domain and use temporal kernel sizes of $[5, 1, 1]$ for the three layers.

## I   Profiling Setup

All latencies refer to training latencies. All models were run on TPUv3 [24] with `bfloat16` precision in TensorFlow 1.x. TPU latencies are measured on 8 TPUv3 cores with a batch size of 1024 (i.e. 128 per core) which is divided by 2 until it fits onto the accelerator's memory. In the cases where a smaller batch size is employed, we normalize the reported latency to the original batch size of 1024 images. For GPU profiling we use a single Tesla-V100 with `float32` precision with a starting batch size of 128, also divided by multiples of 2 if necessary.