# OpenReview forum: "Revisiting ResNets: Improved Training and Scaling Strategies"
_NeurIPS.cc/2021/Conference — NeurIPS 2021 Spotlight_

### Official Review · Reviewer_DbhL · 2021-07-04

**Rating:** 6
**Confidence:** 5

**Summary:**

This paper revisits the performance of ResNet from three aspects, architecture, training scheme, and scaling strategy. The author shows that a better training and scaling strategy can give even more improvements than better architecture design. The author conducted many experiments to empirically show what is a good training and scaling strategy and how good it can be.

**Ethical Concerns:**

No ethical concerns.

**Limitations And Societal Impact:**

The author has addressed the limitations and potential negative societal impact of the work.

**Main Review:**

Strength:
1. There are very thorough experiments to validate the effectiveness of better training and scaling strategy.
2. The concluded training and scaling strategy outperforms state-of-the-art methods on different tasks. The empirical results are very impressive.
3. The paper is easy to follow.

Weakness:
I would say this is a good empirical study about how to better train a network. However, I don’t know if it would be interesting to the NeurIPS audience. The followings are some of my concerns:
1. The effectiveness and interplay of different training/regularization methods have been studied before [1]. Although the author may have conducted more thorough empirical evaluations, the conclusions about this are pretty similar. There are no surprising findings here.
2. The author pointed out the importance of scaling strategy, but the importance of scaling strategy has also been studied before [2][3]. The author further proposed two scaling strategies, but they are all based on empirical results on ResNet. There are no (or very few) analyses on these two strategies. I don’t know if they could also work well on different architectures and tasks.
The author claimed depth scaling is better than width scaling for longer epoch regimes because width scaling is subject to overfitting. However, in Fig.3, width scaling seems to work well in Reso-128 and Reso-160, but fails in Reso-224 and 320. But I think a smaller input should lead to more overfitting, why width scaling works well in small resolutions.
3. The author claimed width scaling could largely increase the number of parameters and lead to overfitting, but we can control the width scaling factor (or even non-uniform width scaling) to make them have the same number of parameters as depth scaling, will the width scaling work better or as good as depth scaling if they have the same number of parameters?
4. In practice, how can we determine the overfitting occurs, or how long can be termed as the long epoch regime to switch from width scaling to depth scaling in different architectures and tasks. If we need to do this empirical evaluation for every architecture and task, I don’t think it is practical.
Overall, I think the results are impressive based on the evaluated architectures, but it lacks further analyses about the results. I don’t know if the proposed scaling strategies can really be useful for practitioners.

Reference

[1] Bag of Tricks for Image Classification with Convolutional Neural Networks. CVPR 2019.
[2] EfficientNet: Rethinking Model Scaling for Convolutional Neural Networks. ICML 2019.
[3] MutualNet: Adaptive ConvNet via Mutual Learning from Network Width and Resolution. ECCV 2020.

**Time Spent Reviewing:**

2

---

> ### Author Response · Authors · 2021-08-08
> **Author response**
>
> We thank the reviewer for thoughtful and constructive feedback. See our replies below.
>
> **“1) Effectiveness and interplay of different training/regularization methods have been studied before. Although the author may have conducted more thorough empirical evaluations, the conclusions about this are pretty similar. There are no surprising findings here”**
>
> We indeed conduct more thorough empirical evaluations and notably point out the interplay between weight decay and other regularization methods, which is missed in prior works. While prior works such as [1] already remark the importance of improved training methods, they do not discuss scaling. In contrast, _we simultaneously study training/regularization and scaling_, which is key in obtaining competitive results on the ImageNet benchmark.
>
> In doing so, we find that _(a) training and scaling strategies may matter more than architectural changes and that (b) ResNets are close to SOTA (and in particular are significantly better than EfficientNets). We also find that (c) supervised learning rivals state-of-the-art self-supervised algorithms._
>
> We believe our results are of interest to the community, since many works routinely incorrectly (i) compare to weak ResNet or supervised baselines and (ii) attribute accuracy gains to architectural changes instead of improved training methods.
>
> **2a) "Importance of scaling strategy has also been studied before (e.g. EfficientNet)”**
>
> Our scaling analysis goes beyond prior works and surfaces common pitfalls in prior research on scaling strategies.
>
> _First, we identify a dependency between the best performing scaling strategy and the training regime_, which is missed in prior works.
>
> _Second, we show that scaling should not be extrapolated from a single (potentially suboptimal) architecture_ as done in EfficientNet which was scaled by extrapolating scaling hyperparameters from EfficientNet-B0.
>
> _We demonstrate empirically that the compound scaling strategy advocated by EfficientNet is suboptimal by designing a faster version of EfficientNet, termed EfficientNet-RS, using our scaling strategy._
>
> **2b) “The author claimed depth scaling is better than width scaling for longer epoch regimes because width scaling is subject to overfitting. However, in Fig.3, width scaling seems to work well in Reso-128 and Reso-160, but fails in Reso-224 and 320. But I think a smaller input should lead to more overfitting, why width scaling works well in small resolutions.”**
>
> As far as we are aware, there is no theoretical/empirical evidence that smaller inputs should lead to more overfitting as claimed by the reviewer. Our experiments empirically show the opposite: width scaling at small resolution does not overfit.
>
> _Finally, Fig 3 shows clearly that depth scaling is preferable to width scaling at all resolutions in longer epoch regimes (irrespective of whether width scaling overfits or not)._
>
> **3) Is width scaling better than depth scaling when controlling for number of parameters?**
>
> In our scaling analysis, ResNet-101 with width multiplier 1.5 and ResNet-300 have a comparable number of parameters. ResNet-300 performs much better (in the long epoch regime) indicating that depth scaling is preferred even when controlling for number of parameters.
>
> We reiterate that number of parameters is not necessarily relevant in practice and we prefer to control for latency as done in Figure 3.
>
> **4) In practice, how can we determine the overfitting occurs, or how long can be termed as the long epoch regime**
>
> Identifying when overfitting occurs goes beyond the scope of our paper. We point out that overfitting can generally be identified by comparing training losses and eval performances for a few different experiments.

---

> > ### Comment · Reviewer_DbhL · 2021-08-23
> > **Reply to author's response**
> >
> > I would like to thank the author for the detailed response. But my concens are not fully addressed. The most important one is that the author didn't address my concern about the generalization ability of the proposed scaling strategy to other structures and tasks. In Table 3, the author just conduct transfer learning to evaluate the effectiveness of the representations. In Table 4, the scaling strategy is not discussed. So I don't know if the empirically-found scaling strategies could hold on other structures and tasks. I see the other reviewers also have similar concerns which are not addressed. I think this is a very important problem since this paper is mainly about empirical evaluation and the strategies are drawn from empirical results. I think a more comprehensive evaluation is necessary. Therefore, I am still leaning towards rejecting this paper.

---

> > > ### Author Response · Authors · 2021-08-26
> > > **Further reply**
> > >
> > > _We addressed all 4 of your primary points above, and since you did not respond to these points, we hope this means that you consider these points addressed. Please let us know otherwise._
> > >
> > > “A more comprehensive evaluation is necessary [...] since strategies are drawn from empirical results.”
> > >
> > > **It seems that we are held to a much higher generalization standard than prior works based on the argument that our strategies are obtained empirically rather than being theoretically motivated.**
> > >
> > > **We respectfully disagree that this should be the case since our scaling strategies readily generalize better than prior works.**
> > >
> > > - **Generalization across model scale**: Through more comprehensive evaluation compared to prior works, we show that previously proposed principled scaling approaches fail to extrapolate. For example, the compound scaling rule from EfficientNet extrapolates scaling coefficients that are optimal (in FLOPs) for small architectures. Our experiments demonstrate that this approach fails to generalize.
> > > - **Generalization across architectures**: ResNets and EfficientNets are representative of most ConvNets (i.e. architectures based on dense convolution or depthwise convolution (e.g. MobileNets)). By demonstrating positive scaling results on both ResNets and EfficientNets, our experiments therefore cover a large part of convolution-based architectures.
> > > - **Generalization across tasks**:
> > >   - _In further work, we observe similar scaling behavior in video classification: scaling the depth with the input resolution performs best for moderate dataset sizes_. In particular, 3D-ResNet-RS-200 obtains competitive top-1 accuracy on Kinetics-400 and 600 (81.0%, 83.8% respectively). Properly interpreting these results requires more context for fairly comparing to baselines and accounting for the additional time dimension. _Given the page limit and our already numerous empirical results, we leave further study of training/scaling in video classification to a paper that will appear soon._
> > >   - _We extensively study scaling as a function of the training regime (e.g. overfitting, dataset size). While this certainly doesn't guarantee strong generalization across tasks, this at least covers the impact of dataset size - which is a main characteristic of a dataset/task._ For example, the scaling strategy advocated by RegNet is based on results obtained when training networks for 10 epochs only and we show that this suboptimal when overfitting is a concern.
> > >
> > > **Further, our paper makes no grand generalization claim - the title specifies clearly that the work is focused on ResNets. Deriving scaling principles that apply across all tasks, modalities and model classes is well outside the scope of the paper.**
> > >
> > > Works which exclusively experiment with image classification regularly get published into top-tier conferences (e.g. EfficientNetv2, NFNet, SimCLRv2). In comparison, we span a much wider range of experimental setups including image classification, video classification and downstream detection/segmentation.
> > >
> > > Finally, our paper is not exclusively dedicated to scaling. We also study the impact of improved training on other tasks (video classification, transferred downstream tasks) and provide fair comparisons against self-supervised baselines.

---

> > > > ### Comment · Reviewer_DbhL · 2021-08-26
> > > > **Reply to author's response2**
> > > >
> > > > Thanks for the detailed response.
> > > >
> > > > 1. I would like to say I am not trying to raise the bar. I just think the generalization issue is important here because the proposed scaling strategy is based on some empirical evaluation on ResNet and EfficientNet. If the proposed strategy cannot really work on other structures/tasks, then I don't think it would be an interesting contribution.
> > > >
> > > > 2. If the author already have some supporting results on video classification, I think putting them in this paper could be a great plus.
> > > >
> > > > 3. I don't think 'some works got accepted' could be a good argument for accepting this paper.
> > > >
> > > > As stated in my original review, I think this is a good empirical study on ResNet. If other reviewers all agree on acceptance, I would be totally fine. But I still suggest the author to conduct more evaluations on other strtuctures/tasks.

---

### Official Review · Reviewer_jadX · 2021-07-16

**Rating:** 6
**Confidence:** 4

**Summary:**

The paper proposes a methodology to improve ResNets like architecture by changing a little bit the architecture and using better training procedure. The rules proposed in the paper allow to have a better trade off between speed and accuracy with ResNet-like architecture and be competitive with efficientNet.

**Limitations And Societal Impact:**

Yes

**Main Review:**

Strength:

- The point "Novel architectures underlie many advances, but are often simultaneously introduced with other critical – and less publicized – changes in the details of the training methodology and hyperparameters." L22 is very interesting it is true that it would be beneficial for the field of image classification if it was more taken into account.
- The paper offers a broad and instructive analysis of different elements of training and how to scale models with resolution.
- The paper is well written and the ideas are easy to understand.

Weaknesses:
- Significance of the results: It would be interesting to also have the results on ImageNet v2 or on another dataset to see how the different results of the different analyses evolve.

- Related work: The EfficientNetv2 paper[1] also proposes to make changes to the training procedure and some architectural changes to have more efficient models. It would be interesting to discuss this paper.

Comment:
- "a canonical ResNet with 79.0%" L31 It would be interesting to specify which type of ResNet, since it is not a ResNet-50, that can be a bit confusing.

[1] Le et al., EfficientNetV2: Smaller Models and Faster Training

**Time Spent Reviewing:**

2

---

> ### Author Response · Authors · 2021-08-08
> **Author response**
>
> We thank the reviewer for thoughtful and constructive feedback. See our replies below.
>
> **Significance of the results / results on another dataset**
>
> We demonstrate the impact of improved training methods on ImageNet classification, Kinetics-400 action recognition as well as transfer performance on Pascal detection/segmentation, ADE segmentation and NYU depth.
>
>
>
> **It would be interesting to discuss EfficientNetv2**
>
> Please note that EfficientNetv2 is concurrent work. While the architecture/method is named EfficientNetv2, it shares very few common characteristics with EfficientNet which can be characterized as the compound scaling method and depthwise convolutions/inverted bottlenecks.
> Furthermore, EfficientNetv2 uses additional regularization tricks (e.g. mixup) as well as evaluation tricks (e.g. reporting training time at a certain resolution and accuracy at a higher resolution). This again illustrates the importance of disentangling the architecture from regularization/training/scaling, which is a central point of our paper. We will discuss EfficientNetv2 in the final draft.
>
>
>
> **“a canonical ResNet with 79.0%"**
>
> Good point, we will specify that this is a ResNet-200 in the final draft.

---

> > ### Comment · Reviewer_jadX · 2021-08-31
> > **Thanks for your response**
> >
> > Thanks for your response. The rebuttal have answered most of my concerns. So I will keep my initial score which is leaning towards accepting the paper.

---

### Official Review · Reviewer_trei · 2021-07-17

**Rating:** 8
**Confidence:** 4

**Summary:**

This paper presents an in-depth analysis of how improvements in training, regularization, and model architecture affect the performance and efficiency of ResNet models. Combining all the improvements, the proposed ResNet-RS models achieve significant speedup over EfficientNet (a strong baseline).

**Limitations And Societal Impact:**

1. The scaling rule in Appendix D Table 9 seems to be arbitrary to me which makes the scaling method hard to be applied to other architecture. In the text, the authors say that they try to keep the ratio roughly constant which contradicts what Table 9 presents: 24:36 = 1:1.5 while 29:53 ~= 1:1.8, 36:72 = 1:2, and 44:87 ~= 1:1.97. Only the last two models have a similar ratio. Moreover, I understand that the authors try to keep the small models (ResNet-50, ResNet-101, ResNet-152) the same as the prior work, but I wonder whether these choices are optimal. Would it be possible to find some small ResNets which push the Pareto curve further?
2. The authors do not study how the improvements in three directions generalize to other model architectures and datasets. I understand that the authors focus on ResNet architecture, but the paper would be more impactful if the authors can provide a study on the impact of these methods on different model architectures. Given that EfficientNet-RS is worse than ResNet-RS, I believe the ranking of model architectures can be shuffled.


**Main Review:**

Strengths:
1. While the methods of training neural networks used in this paper are not novel, the authors provide a thorough and original analysis of existing methods.
2. This paper proposes a novel model scaling strategy: 1. scaling the depth when overfitting occurs 2. scaling the image resolution slower.
3. The writing is clear and well-written.
4. This paper provides several interesting observations and insights.
5. This paper shows that the choice of regularization can be affected by the number of training epochs which suggests that some models searched with small training budgets (such as RegNet) may not generalize to longer training procedures.
6. The authors show that with improvements in training, regularization, and model architecture, simple ResNets can outperform strong EfficientNets. Further, in Figure 5, the authors show that ResNet-RS outperforms EfficientNet-RS, which suggests that the EfficientNet architecture is also worse under a fairer comparison.
7. The authors also provide a comparison with the unsupervised method and show that the proposed model also achieves better fine-tuning results.
8. Not limited to ImageNet, the authors also show that the proposed method can be applied to 3D ResNet for Video Classification.

Weaknesses:
1. The text in Appendix B1 is missing. I can't find any text describing Figure 5 and the comparison with EfficientNet-RS.
2. Scaling the image resolution slower has been proposed in the fast compound scaling method [1]. While other parts are different, I wonder how the proposed scaling strategy performs when compared with the fast compound scaling in terms of performance-efficiency trade-off.

Questions:
1. In table 4, the EMA of weights does not lead to any performance improvement. Does it also happen to the final model? If it is unnecessary, why don't you just remove it?
2. I wonder how general the scaling strategy is. Is it only limited to vision models or it generalizes to NLP and speech? Or is it only limited to ConvNets?
Can it also be applied to vision transformers [2] or their variants?


Overall, this is a well-written paper and provides several new observations and insights. I view the lack of novelty in the training methods and model design as a strength of this paper instead of a weakness. Undoubtedly, I recommend accepting it. While I give it an 8 currently, I am willing to raise my rating to 9 if the authors can address the limitations mentioned below or even to 10 if my questions are addressed as well.

References:
[1] Dollár, Piotr, Mannat Singh, and Ross Girshick. "Fast and accurate model scaling." Proceedings of the IEEE/CVF Conference on Computer Vision and Pattern Recognition. 2021.
[2] Dosovitskiy, Alexey, et al. "An image is worth 16x16 words: Transformers for image recognition at scale." arXiv preprint arXiv:2010.11929 (2020).

**Time Spent Reviewing:**

4

---

> ### Author Response · Authors · 2021-08-08
> **Author response**
>
> We thank the reviewer for thoughtful and constructive feedback. See our replies below.
> ---
>
> ---
> ### Limitations
>
> **Scaling rule in Table doesn’t show constant ratios between c3 and c4. Are the original choices for ResNets optimal**
>
> - The newly introduced larger ResNets have ratios close to 1:2.
> We believe that ResNet-270 with 28:54 (instead of 29:53) would have similar results.
>
> - 24:36 (1:1.5) corresponds to the already existing ResNet-200.
> Although probably not optimal, we found smaller ResNets (from 50 to 200) to have very competitive configurations.
>
> **Generalization to other architectures and datasets?**
>
> - We show that the scaling strategy generalizes to other architectures by designing EfficientNet-RS, an improved version of EfficientNet, using our proposed scaling strategies.
>
> - We show the impact of improved training strategies on Kinetics-400 as well as transfer performance on different datasets (Pascal detection/segmentation, ADE segmentation and NYU depth)
>
> ---
>
>
> ---
> ### Weaknesses
> **Missing text in Appendix B1 and Figure 5**
>
> Appendix B1 actually only consists of Figure 5 which has a self-descriptive caption.
> We will add some text in the final draft to clear the confusion.
>
> **Comparison to fast compound scaling?**
>
> Fast compound scaling (as introduced in concurrent work [1]) encourages primarily scaling model width, while scaling depth and resolution to a lesser extent.
>
> This matches our experimental results which show that width scaling (close to fast compound scaling) outperforms depth scaling in the low epoch training where overfitting is not an issue. We expect fast compound scaling from [1] to perform worse in regimes where overfitting occurs as suggested by our experiments.
>
> ---
>
>
> ---
> ### Questions
> **Is EMA useful for the final models? If not, why not remove it?**
>
> We’ve found EMA to be only marginally useful in the final models.
> We included it mostly to match the EfficientNet’s paper experimental setup.
> Using EMA has the benefit of smoothing learning curves which can help in practice.
>
> **Does the scaling strategy generalize to NLP/speech and Transformers?**
>
> This is a very interesting question which goes beyond the scope of our paper.
> We hypothesize that the width vs depth scaling analysis and its dependency to the training regime might generalize to other domains such as NLP and speech.

---

> > ### Comment · Reviewer_trei · 2021-08-16
> > **Reply to authors' responses**
> >
> > The authors have answered some of my questions. However, my main concern of whether the proposed method generalizes to other models (beyond ResNets and EfficientNets) or more datasets (not just the two in the paper) remains not addressed. Therefore, I would just keep my rating.

---

### Decision · Program_Chairs · 2021-09-27

**Decision:**

Accept (Spotlight)

**Comment:**

All reviewers unanimously recommend acceptance of the paper. The paper is well written and instructive about the importance of training and scaling strategies. It may deserve some elevated attention as a spotlight contribution.